# Hippocampal projections to the anterior olfactory nucleus differentially convey spatiotemporal information during episodic odour memory

Afif J. Aqrabawi [1] & Jun Chul Kim[1,2]

The hippocampus is essential for representing spatiotemporal context and establishing its association with the sensory details of daily life to form episodic memories. The olfactory cortex in particular shares exclusive anatomical connections with the hippocampus as a result of their common evolutionary history. Here we selectively inhibit hippocampal projections to the anterior olfactory nucleus (AON) during behavioural tests of contextually cued odour recall. We find that spatial odour memory and temporal odour memory are independently impaired following inhibition of distinct, topographically organized hippocampal-AON pathways. Our results not only reveal a longstanding unknown function for the AON but offer new mechanistic insights regarding the representation of odours in episodic memory.

---

[1] Department of Cell and Systems Biology, University of Toronto, Toronto, Ontario, Canada M5S 3G5. [2] Department of Psychology, University of Toronto, Toronto, Ontario, Canada M5S 3G3. Correspondence and requests for materials should be addressed to J.C.K. (email: kim@psych.utoronto.ca)

What happened, when, and where? The ability to readily integrate elements of a unique event into a single representation is a fundamental property of episodic memory[1]. Encoding, storing, and retrieving contextually unique episodes is crucial for making sense of the present, forming predictions about the immediate future, and selecting behavioural responses relevant to survival[2–4].

Lesion and recording studies in humans and non-human animals have highlighted a central role of the hippocampus (HPC) in mediating episodic memory[5,6]. The spatial and temporal context of an event are first encoded within the HPC as the collective activity of place and time cells[7,8]. Contextual information later serves as a potent retrieval cue, bringing about the rich multisensory details of the original experience[9,10]. An emerging theory holds that the HPC conducts this retrieval process by reinstating patterns of cortical activity observed during learning[11,12]. Yet, it is not known how hippocampal transmission

of contextual information can reproduce the sensory aspects of episodic memory.

Olfaction is considered the most evolutionarily ancient sense as evidenced by the direct anatomical connections between the olfactory cortex and the limbic system[13,14]. In particular, hippocampal projections to the olfactory cortex offer a unique experimental model for understanding the context-driven recollection of previously encountered sensory stimuli. Recently, we revealed a dense and topographically organized projection from the dorsoventral extent of the HPC to the anterior olfactory nucleus (AON)[15] (Fig. 1a). The AON receives unidirectional, monosynaptic inputs from the CA1, in contrast to other primary sensory areas that receive hippocampal inputs indirectly via adjacent medial temporal lobe structures[16].

The AON is an ideal site of convergence for olfactory and contextual information given its anatomical position as the initial recipient of input from the olfactory bulb and the largest source

**Fig. 1** Inhibition of AON-projecting hippocampal neurons impairs context-dependent odour memory recall. **a** Coronal section depicting site of AAV-ChR2-YFP and -mCherry injections in the hippocampus (top) and the resulting innervation pattern at the AON (bottom). Coordinates mark anteroposterior position from bregma. Scale bars represent 1 mm; ac, anterior commissure; pc, piriform cortex; l, lateral AON; m, medial AON; vp, ventroposterior AON. **b** Schematic diagram depicting experimental approach. CAV2-Cre was infused into the AON, whereas Cre-responsive AAV-hM4D-mCherry was injected in the hippocampus. **c** Representative sections depicting AON-projecting HPC neurons expressing hM4D-mCherry. Magnification is indicated on the bottom right of each panel. Scale bars represent 1 mm (black) or 10 μm (white). **d** The olfactory spatial memory test paradigm. CNO-treated hM4D mice investigated the familiar and novel odour location to a largely equal extent, indicative of impaired spatial odour memory (Independent-samples t-test, $t_{(16)} = 3.194$, **$P < 0.01$). **e** The olfactory temporal order memory test paradigm. CNO-treated hM4D mice were impaired in memory for the temporal occurrence of encountered odours (Independent-samples t-test, $t_{(16)} = 2.795$, *$P < 0.05$). **f** Both groups showed normal performance in a context-independent novel odour recognition test (Independent-samples t-test, $t_{(16)} = 0.05644$, NS, $P = 0.9557$). For all behavioural tests, CNO was injected 15 min before the retrieval phase. The odour in the novel spatiotemporal configuration is numbered in red. Positive discrimination ratios indicate preference for the novel odour-context configuration. mCherry control group: $n = 8$, hM4D group: $n = 10$. Data are presented as mean ± SEM

of feedback projections within the olfactory cortex[17,18]. Consistently, it has been shown that hippocampal inputs to the AON can alter olfactory perception and odour-guided behaviours[19]. However, the functional role of the HPC-AON pathway remains unexplored. Here we combine chemogenetic and optogenetic approaches to demonstrate that information regarding the spatial and temporal context of odour memory is delivered by topographically organized hippocampal inputs to the AON.

## Results

**AON-projecting HPC cells support context-dependent odour memory recall.** Using c-Fos as a proxy for neural activity we measured the AON's response to a novel odour and context, separately or in combination. We found that the AON exhibits a widespread increase in c-Fos-positive neurons following exposure to a novel context-odour pairing, yet neither stimulus alone was sufficient to evoke an increase in the number of labelled cells compared with homecage controls (Supplementary Fig. 1). These results are in line with previous work on the response properties of AON principal cells and suggest that the AON has a role in context-odour coincidence detection[20,21].

To manipulate activity in the HPC-AON pathway, we infused the retrogradely propagating canine adenovirus encoding Cre recombinase (CAV2-Cre) into the AON, followed by HPC infusions of an Adeno-associated virus (AAV) vector carrying a Cre-dependent inhibitory hM4D-mCherry (AAV8-hSyn-FLEX-hM4D-mCherry) or mCherry alone (AAV8-hSyn-FLEX-mCherry) (Fig. 1b). This allowed us to selectively inhibit AON-projecting HPC cells upon the administration of clozapine-N-oxide (CNO), thereby limiting the AON's response to a context-odour pairing (Supplementary Fig. 2). Cre-mediated viral mCherry expression was observed throughout the ventral two thirds of the hippocampal CA1 (Fig. 1c). Three weeks after viral infusion, mice underwent behavioural tests to evaluate memory for the associations between odours and the spatiotemporal context in which they occurred. Assessing the retrieval of episodic odour information was made possible by capitalizing on the innate tendency of mice to preferentially investigate novel stimuli[22]. Thus, mice can behaviourally express correct memory by investigating odours paired with a novel position in space, or temporal sequence, more so than familiar configurations.

We first tested the ability to remember where specific odours occurred in context. In two encoding phases of a spatial odour memory test, mice were presented with two different odours (odour 1 and 2) placed on opposite ends of a distinct context (A) for 5 min. Mice were then placed in a separate context (B) where the same odours were positioned in reversed locations. Each exposure was followed by a 15 min retention delay, the second of which was preceded by a CNO injection. In the subsequent retrieval phase, animals were reintroduced to context A wherein two copies of either odour 1 or 2 were presented on both sides. In this paradigm, correct memory expression would drive mice to investigate the odour found at the novel location (NL) within the context, as seen in control mice expressing mCherry alone (Fig. 1d). In contrast, CNO-treated hM4D mice investigated both, otherwise identical odours for a similar proportion of time. These results cannot be explained by differences in total investigation time or distance travelled as both groups displayed similar measures in each (Supplementary Fig. 3).

Next, we examined memory for when specific odours were encountered. Mice were presented with a sequence of odours in three successive encoding phases followed by a retrieval phase where two of the previously encountered odours were reintroduced. Control mice preferentially investigated the odour

encountered earlier in the sequence, yet CNO-treated hM4D mice investigated both odours to a largely equal extent (Fig. 1e). Importantly, the spatial context was consistent throughout the test where novelty was only conferred by temporal distance (TD) between the two odours. Lastly, to examine whether hippocampal function can be extended to memory for odours regardless of context, we conducted a novel odour recognition test where animals were presented with a familiar and previously unexplored odour. Both groups preferentially investigated the novel odour despite CNO administration (Fig. 1f). Discrimination ratios obtained for both groups under all conditions were compared to zero (chance performance) and the results detailed in Supplementary Table 1. Together, our findings indicate that AON-projecting HPC cells mediate the retrieval of odour memory only when it is tied to spatiotemporal context.

**Spatiotemporal information is transmitted along distinct HPC-AON pathways.** The ring of cells which constitute the AON is typically partitioned into four subregions based on their relative cardinal positions: pars medialis (mAON), pars dorsalis (dAON), pars lateralis (lAON), and pars ventralis (vAON)[17,23]. Previous studies have found that each subregion displays distinct cytoarchitectural, morphological, neurochemical, and connectivity patterns, thereby providing critical foundations for defining AON subregional boundaries[23] (for a review, see ref. [17]). Our recent neuroanatomical study revealed a previously unknown topographic gradient in HPC-AON projections such that the ventral-most part of the HPC innervates most heavily the medial aspect of the AON and, progressively, more dorsal parts of the HPC innervate increasingly more lateral positions at the AON[15]. As a result, our definition of the mAON encompasses the pars medialis, yet includes a portion of what is traditionally accepted to be pars dorsalis.

Such topographically organized HPC terminals at the AON may transmit distinct contextual cues depending on where they arise within the HPC. To explore this idea, we employed archaerhodopsin (ArchT) to optogenetically inhibit intermediate HPC (iHPC) or ventral HPC (vHPC) terminals that innervate separate AON subregions. One group of mice received bilateral infusions of AAV-CaMKIIa-ArchT-eYFP into the iHPC with optic fibre implantations in the lateral AON (lAON) for inhibiting the iHPC-lAON pathway and another group underwent viral infusions into the vHPC with optic fibre implantations in the medial AON (mAON) for inhibiting the vHPC-mAON pathway (Fig. 2a; Supplementary Fig. 4). A control group given counterbalanced infusions and implantations were prepared using AAV expressing green fluorescent protein (GFP) only. The efficacy of optogenetic inhibition was confirmed in a separate group of mice unilaterally infused with ArchT, where hippocampal terminal inhibition during a novel context-odour encounter selectively reduced the density of c-Fos positive cells within the AON ipsilateral to the manipulation, leaving the contralateral side unaffected (Supplementary Fig. 5).

Animals were tested in the same behavioural paradigms used for hM4D experiments, except inhibition was mediated by light (532 nm, 12 mW) limited to the retrieval phase. Illumination of the iHPC-lAON and vHPC-mAON pathways disrupted memory for where odours were encountered, as both groups were unable to discriminate odours tied to a spatial location within a specific context (Fig. 2b). In contrast, only vHPC-mAON, but not iHPC-lAON pathway inhibition impaired memory for when odours occurred in a temporal sequence (Fig. 2c). Importantly, all groups showed similar levels of novelty preference in a context-independent novel odour recognition test (Fig. 2d). A comparison of the discrimination ratios with chance performance can be

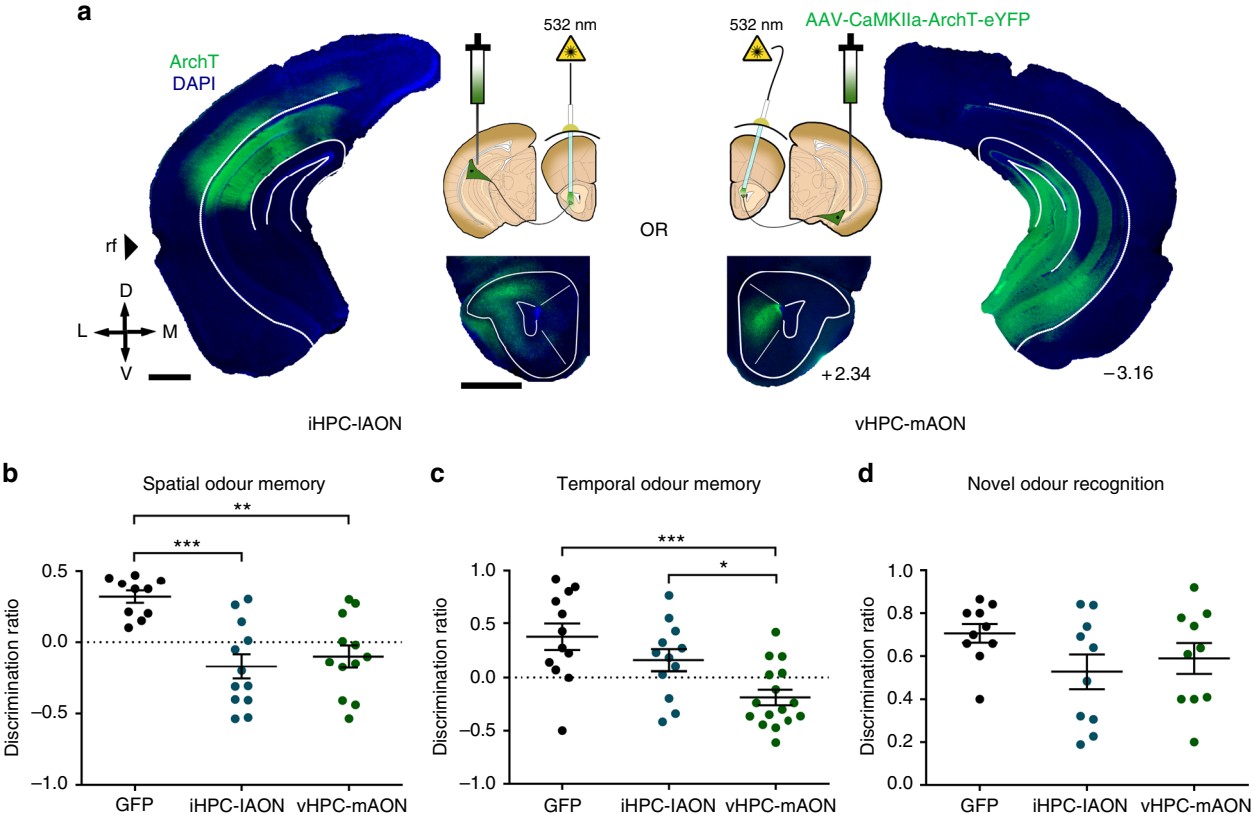

**Fig. 2** Spatial and temporal information is transmitted along separate HPC-AON circuit elements. **a** Diagram depicting location of ArchT infusions within the HPC and optic fibre implantations above the AON in both experimental groups. iHPC terminals are predominantly found in the lAON (left) whereas vHPC fibres preferentially innervate the mAON (right). **b** Inhibition of both iHPC-lAON and vHPC-mAON pathways impaired spatial odour memory (GFP: $n = 10$; iHPC-lAON: $n = 12$; vHPC-mAON: $n = 12$; one-way ANOVA, $F_{(2,31)} = 11.65$, ***$P < 0.0005$). **c** Inhibition of the vHPC-mAON, but not iHPC-lAON pathway impaired memory for the temporal order of a sequence of odours (GFP: $n = 12$; iHPC-lAON: $n = 12$; vHPC-mAON: $n = 16$; one-way ANOVA, $F_{(2,37)} = 9.235$, ***$P < 0.005$). **d** HPC-AON pathway is not necessary for context-independent novel odour recognition (all groups: $n = 10$; one-way ANOVA, $F_{(2,27)} = 1.824$, NS, $P = 0.1807$). Scale bars represent 1 mm. Coordinates indicate anteroposterior position from bregma. Data are presented as mean ± SEM

found as Supplementary Table 2. The identified impairments in episodic odour memory retrieval suggest that representations of space and time are differentially distributed across the AON whereby both iHPC and vHPC inputs deliver spatial information, but information regarding the temporal context is supported only by vHPC inputs.

The behavioural tests employed thus far were designed to separately examine the spatial and temporal elements of episodic odour memory. However, given the unified nature of recollected episodes, it is necessary to probe memory-based behavioural judgements while the animal concurrently weighs both parameters. To investigate further the spatiotemporal contributions of hippocampal inputs to the AON, we adopted an episodic memory test where recollection of an odour, its spatial location, and temporal occurrence (what–when–where) were tested simultaneously[24,25]. The test involved two encoding phases and one retrieval phase, each separated by a 1 h delay (Fig. 3a). Light-mediated inhibition was limited to the retrieval phase. During the encoding phases, mice sampled two distinct odours placed at adjacent corners of an open field, and then sampled a new set of odours positioned on the opposite side of the arena. In the retrieval phase, all four odours were presented, each possessing a unique spatiotemporal configuration. Encountered odours were either in a novel location and appeared earlier in temporal distance (NL/TD), familiar in location/temporally distant (FL/TD), novel in location/temporally recent (NL/TR), or familiar in location/temporally recent (FL/TR).

The control group displayed a pattern of investigation indicative of their novelty preference such that the greatest proportion of time was spent investigating the odour encountered in the most novel spatiotemporal combination (NL/TD), while investigation time for the remaining odours decreased in a familiarity-dependent manner: (NL/TD) > (FL/TD) ≃ (NL/TR) > (FL/TR) (Fig. 3a, b). The iHPC-lAON group spent the greatest proportion (~50%) of time investigating the odour with the FL/TD configuration, while the NL/TR odour was investigated least. Strikingly, inhibition of the vHPC-mAON pathway produced an inverse pattern of investigation to the control group. The odour presented in the FL/TR configuration was investigated most, while the NL/TD odour was investigated least. A separate analysis delineating the spatial and temporal contributions to memory revealed that iHPC-lAON pathway inhibition impaired spatial but not temporal odour associations, while the vHPC-mAON pathway inhibition impaired both components to a similar extent (Fig. 3c). Collectively, these findings confirm the differential roles of the iHPC-lAON and vHPC-mAON pathways in conveying spatiotemporal information during olfactory episodic memory retrieval.

**HPC-AON input is necessary for context-driven odour recall.** The ability to explicitly recollect sensory information is a hallmark of episodic memory, particularly when lacking a physical sensory cue[26]. Context alone can drive the activity of primary

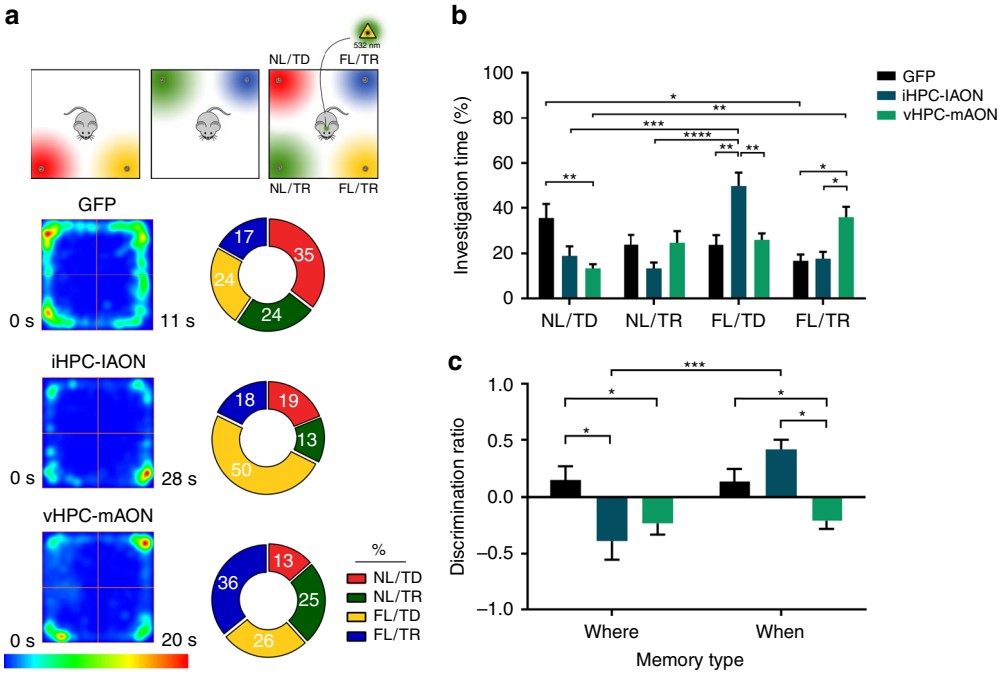

**Fig. 3** HPC-AON pathways contribute distinct spatiotemporal information during episodic odour recall. **a** Top: illustration of the episodic memory test paradigm. Bottom: heat maps depicting group average position (left) and pie charts representing the percent investigation time (right) during the retrieval phase of the episodic memory test (all groups: $n = 12$). **b** Comparison of percent investigation time for all groups during the retrieval phase of the episodic memory test (two-way ANOVA, treatment-by-odour interaction $F_{(6,132)} = 9.532$, ****$P < 0.0001$, main effect of odour $F_{(3,132)} = 5.415$, **$P < 0.005$). **c** iHPC-lAON pathway inhibition impaired the spatial, but not temporal component of odour memory, while vHPC-mAON pathway inhibition impaired both components to a similar extent (two-way ANOVA treatment-by-memory type interaction $F_{(2,66)} = 9.320$, ***$P < 0.0005$; main effect of treatment $F_{(2,66)} = 6.715$, **$P < 0.005$; main effect of memory type $F_{(1,66)} = 9.620$, **$P < 0.005$). FL/TD, familiar location/temporally distant; FL/TR, familiar location/temporally recent; NL/TD, novel location/temporally distant; NL/TR, novel location/temporally recent. Data are presented as mean ± SEM

olfactory regions to form internal representations of odours although the underlying neural circuit is unknown[9,10]. To this end, we examined whether the HPC-AON pathway could support this function. Mice were allowed to explore a rich spatial context in the presence of a pure odour emitted from a cotton swab tip for 30 min per day over nine consecutive days. On day 10, mice were reintroduced to the context in the absence of the applied odour. The mismatch between the odour-paired context and the lack of an emitted scent drives an increase in the investigation time for the cotton swab (Supplementary Movie 1). Indeed, control mice exhibited a marked increase in investigation of the cotton swab upon failure to find the expected odour (Fig. 4a; Supplementary Fig. 6). In contrast, hM4D- and ArchT-mediated inhibition of the HPC-AON pathway abolished this behaviour. All groups showed no increase in investigation time when similarly trained but tested in a novel context (Fig. 4b). In a separate group of mice trained on an odour-context association and later exposed to the context alone, immunostaining for c-Fos revealed a context-driven activation of the AON, complimentary to similar activity patterns observed in the olfactory bulb and piriform cortex (Supplementary Fig. 7)[9,10]. Together, these results indicate that HPC-AON communication is necessary for mediating the context-driven recollection of odours.

## Discussion

The HPC has an essential role in organizing memory for sensory events within the framework of spatiotemporal context, a function integral to episodic memory[27]. Memory for sensory experiences are theorized to be distributed in modality-specific neocortical networks[12,26,28–31]. The scattered components of the engram are bound by the HPC which acts as an integrative hub,

providing storage for context-based associative indices[32]. Later during retrieval, the HPC uses the associative index to reinstate patterns of cortical activity that were observed during encoding[11]. One functional magnetic resonance imaging study in particular provided initial evidence for the reactivation of the olfactory cortex during odour memory recollection[9]. In their study, human subjects were instructed to learn associations between an odour and a context after which they were examined for the effect of the context alone on neural activity during memory retrieval. Both the primary olfactory cortex and the anterior HPC were activated, adding further support for the role of the HPC in binding sensory memory traces preserved in modality-specific neural regions.

We show that HPC inputs to the AON are necessary for mediating the context-driven recollection of odours (Fig. 4). In our context-driven odour recollection paradigm, the context alone has been shown to produce an internal representation of the associated odour and drive behavioural investigation of the empty odour source[10]. Whether the activity of place and time cells within the HPC are themselves sufficient to trigger retrieval of the odour memory remains unknown. It is possible that the increased firing rate of 'misplace' or mismatch cells within the HPC facilitates the reinstatement of odour representations within the AON, particularly when mice failed to find the odour paired with the context. These hippocampal principal neurons were first characterized in O'Keefe's[33] landmark paper where place units were also described. Misplace cells fire maximally after encountering novel stimuli or at the mismatch between experience and expectation (i.e., when rats sniffed in a particular location where a novel stimulus was found or when rats failed to find the expected stimulus which had been there)[33]. Thus, the firing of misplace cells may serve as a signal to instigate odour recollection

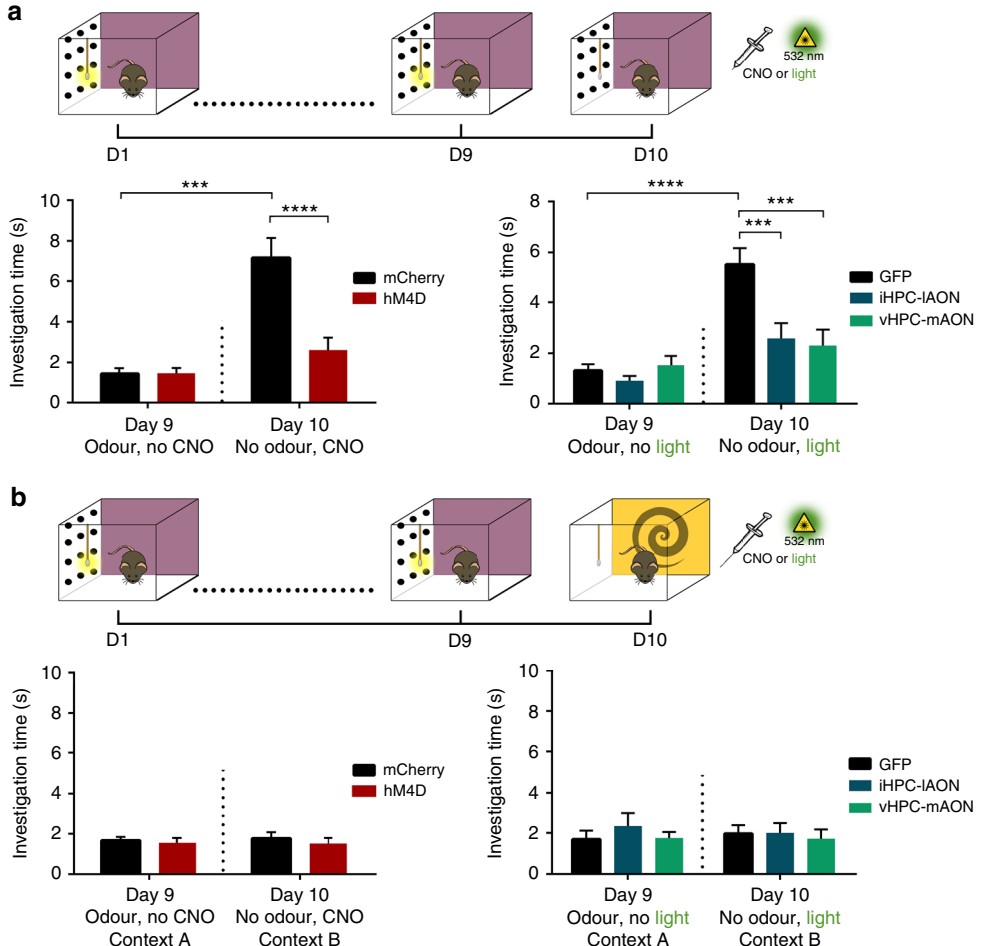

**Fig. 4** The HPC-AON circuit is necessary for context-driven odour recall. **a** Top: contextually cued odour recall test paradigm. Bottom: control groups in both experiments showed an increased investigation time between Day 9 and 10. Yet, following hM4D- (left) or ArchT-mediated (right) inhibition of hippocampal projections to the AON, mice fail to exhibit this behaviour (hM4D experiment- hM4D-mCherry: $n = 10$, mCherry control: $n = 8$; two-way ANOVA main effect of treatment $F_{(1,32)} = 14.85$, ***$P < 0.0005$; main effect of Day $F_{(1,32)} = 34.5$, $P < 0.0001$; interaction between treatment and Day $F_{(1,32)} = 15.19$, ***$P < 0.0005$; ArchT experiment—GFP: $n = 14$, iHPC-lAON: $n = 12$, vHPC-mAON: $n = 12$; two-way ANOVA main effect of treatment $F_{(2,70)} = 7.538$, **$P < 0.005$; main effect of Day $F_{(1,70)} = 31.45$, ****$P < 0.0001$; interaction between treatment and Day $F_{(2,70)} = 7.052$, **$P < 0.005$). **b** Mice were trained on a context-odour association and tested in a distinct context B in the absence of an applied odour. All animals showed no increase in investigation of the cotton swab despite all aspects of training being equivalent to mice tested in the training context (hM4D experiment—hM4D-mCherry: $n = 10$, mCherry control: $n = 8$; two-way ANOVA main effect of treatment $F_{(1,32)} = 0.5035$, NS, $P = 0.4831$; main effect of Day $F_{(1,32)} = 0.0581$, NS, $P = 0.8111$; interaction between treatment and Day $F_{(1,32)} = 0.01156$, NS, $P = 0.9150$; ArchT experiment—GFP: $n = 12$, iHPC-lAON: $n = 12$, vHPC-mAON: $n = 12$; two-way ANOVA main effect of treatment $F_{(2,66)} = 0.4907$, NS, $P = 0.6144$; main effect of Day $F_{(1,66)} = 0.007502$, NS, $P = 0.9312$; interaction between treatment and Day $F_{(2,66)} = 0.2189$, NS, $P = 0.8040$). Data are presented as mean ± SEM

when the odour is no longer present. Inhibition of the HPC-AON pathway may have interrupted the downstream propagation of this signal thereby preventing the exploratory behaviour usually displayed.

To our knowledge, this work provides the first functional demonstration that hippocampal inputs to the olfactory cortex are necessary for episodic odour memory. Specifically, we show that inhibition of hippocampal inputs to the AON results in a loss of odour memory when it is tied to spatiotemporal context (Fig. 1). Next, we demonstrated that hippocampal representations of space and time are differentially distributed across the AON such that both iHPC and vHPC inputs transmit spatial information, whereas information regarding the temporal context is supported by vHPC inputs (Figs. 2, 3). The segregation of hippocampal inputs by their spatiotemporal contribution is consistent with recording studies which suggest that each dimension of an event is coded independently within overlapping

hippocampal neural populations[34]. Although the dorsal HPC is commonly thought to dominate spatial processing, particularly when contrasted to the ventral aspect, this dichotomic perspective is being revised in light of growing evidence for a hippocampal model characterized by multiple levels of functional long-axis gradients and discrete domains[35]. This alternative perspective better accommodates recent findings demonstrating iHPC and vHPC involvement in transmitting spatial information and is in accordance with our results[24,36]. Lastly, in addition to the AON, HPC axon fibres were found, albeit to a much lesser extent, within the tenia tecta, an extension of the HPC into the olfactory peduncle. Little is known about the function of this structure, thus the contribution of HPC inputs to the tenia tecta remains undetermined.

Ultimately, our findings support a model of episodic odour memory whereby information regarding odour quality and spatiotemporal context merge at the level of the AON and, as a

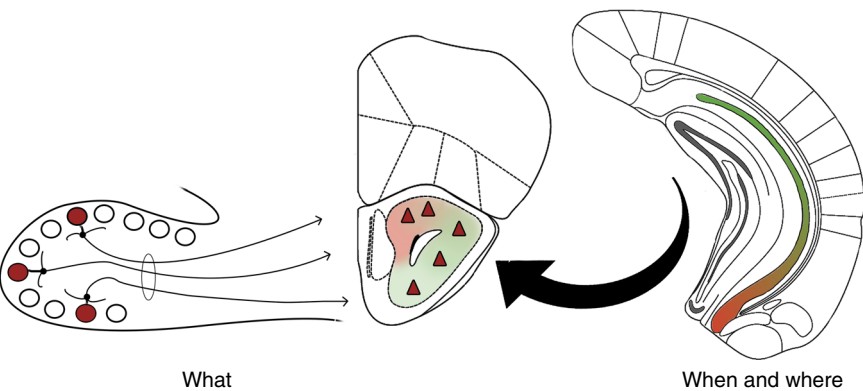

What                                When and where

**Fig. 5** Conceptual diagram for the formation of episodic odour memory. A model of episodic odour memory whereby information regarding odour quality and spatiotemporal context merge at the level of the AON producing cellular populations that represent previously encountered odours (what) within the context in which they occurred (when and where)

natural consequence of Hebbian synaptic plasticity, produce cellular populations that represent previously encountered odours within the context in which they occurred (Fig. 5). Such a system maintains the fidelity of the original memory and allows access for retrieval of the full trace via partial cues from either olfactory or contextual inputs.

Our findings reveal a previously unreported function for the AON, a structure that has long remained elusive in its role in olfactory information processing. The AON, in its position immediately posterior to the olfactory bulb and anterior to the piriform cortex, maintains connections at nearly every synaptic step in the olfactory pathway[17]. Its central arrangement and extensive command over activity within the olfactory cortex makes the structure a conceivable repository of episodic odour engrams. Adding further weight to this possibility is the AON's well-documented involvement in the early pathophysiology of Alzheimer's disease, a debilitating disorder of episodic memory[37–41]. In fact, the AON has repeatedly been identified as the earliest site of neurodegeneration, including cell loss and the formation of neurofibrillary tangles, senile plaques, and other associated histological markers[42–45]. Complementing these observations is a vast body of work which reports olfactory dysfunction, specifically olfactory memory loss and hyposmia, as symptoms of the onset of Alzheimer's disease[46–49]. Consistently, our previous findings have also implicated the AON in the top-down modulation of olfactory sensitivity[19]. Studies of olfactory deficits associated with Alzheimer's disease have now prompted measurements of olfactory ability to be used as clinical bio-markers in the early diagnosis of the disease[50–52].

Our findings implicate the AON as the storehouse of episodic odour engrams and suggest that its accelerated degeneration may underlie the olfactory dysfunction observed in Alzheimer's disease. Thus, the HPC-AON pathway provides a novel circuit model for studying fundamental aspects of human episodic memory and the odour memory deficits commonly found in neurodegenerative conditions.

## Methods
**Animals**. Male C57BL/6 mice (Charles River Laboratories) were used in all behavioural tests. All mice were 8–10 weeks old at the time of surgery and 12–14 weeks old at the time behavioural testing began. Before surgery, mice were group-housed in a temperature-controlled room on a 12 h light/dark cycle with ad libitum access to food and water. Following surgery, mice were individually housed. A total of 60 mice were distributed into two groups for hM4D experiments (hM4D-mCherry: $n = 10$, mCherry control: $n = 8$) and three groups for ArchT experiments (GFP-control: $n = 14$, iHPC-lAON: $n = 12$, vHPC-mAON: $n = 16$). An additional 29 animals were used for c-Fos immunoreactivity experiments. All procedures were performed in accordance with the guidelines of the Canadian Council on Animal Care and the University of Toronto Animal Care Committee.

**Surgical procedures**. CAV2-Cre viral vector was purchased from the Plateforme de Vectorologie de Montpellier, AAV2/8-hSyn-FLEX-hM4D-mCherry (hM4D) and AAV2/8-hSyn-DIO-mCherry (mCherry control) from the vector core at University of North Carolina at Chapel Hill, and AAV2/5-hSyn-hChR2-mCherry (ChR2-mCherry), AAV2/5-hSyn-hChR2-eYFP (ChR2-YFP), AAV2/5-CaMKIIa-ArchT-eYFP (ArchT), and AAV2/8-CB7-CI-EGFP-RBG (GFP-control) from the University of Pennsylvania Vector Core. Stereotaxic surgery was conducted on mice anaesthetized with isoflurane and administered ketoprofen (5 mg/kg) for pain management. For chemogenetic experiments, CAV2-Cre viral vector was bilaterally infused into the mAON (10° angle toward midline; anterior/posterior (AP): + 2.90, medial/lateral (ML): ± 1.10, dorsal/ventral (DV): − 3.42) and lAON (no angle; AP: + 3.20, ML: ± 1.10, DV: − 3.90) at a volume of 0.1–0.2 μL and hM4D into the iHPC (no angle; AP: − 2.70, ML: ± 2.20, DV: − 2.00) and vHPC (10° angle away from midline; AP: − 2.92, ML: ± 2.15, DV: − 4.90) at a volume of 0.3–0.4 μL. The position immediate to the rhinal fissure was used as a landmark to delineate intermediate and ventral parts. For optogenetic experiments, ArchT or GFP-control was bilaterally infused into the iHPC or vHPC in a volume of 0.3–0.4 μL, and optical fibres (200 μm core diameter, 0.39 NA; Thorlabs, Newton, NJ, USA) threaded through 1.25 mm-wide zirconia ferrules (Thorlabs) were bilaterally implanted into the lAON or mAON, respectively. For anterograde tracing experiments, ChR2-mCherry was infused into the vHPC and ChR2-YFP was infused into the contralateral iHPC. All infusions were made by means of pressure ejection at a rate of 0.1 μL/min through a cannula connected by Tygon tubing to a 10 μL Hamilton syringe (Hamilton, Reno, NV). A 15 min interval was allotted after each infusion to limit the viral spread.

**Drugs**. CNO obtained from the NIH was dissolved in a solution of 10% dimethyl sulphoxide and 0.9% saline. A dose of 5 mg/kg of CNO was used in all behavioural experiments using hM4D-mCherry- and mCherry-only-expressing animals. All CNO treatments were separated by a minimum of 72 h.

**Apparatus for optogenetic experiments**. Inhibition of hippocampal terminals at the AON was conducted by illumination with green light (532 nm, 12 mW) generated by a diode-pumped solid state laser (Laserglow, Toronto, ON, Canada). The laser was connected to a 1 × 2 optical commutator (Doric Lenses, Quebec, QC, Canada), which divided the light path into two arena patch cables attached to the implanted optical fibres.

**Experimental design**. Unless otherwise noted, all tests took place in a 50 cm × 25 cm × 20 cm plexiglass open-topped cage. Odours were presented mixed with woodchip bedding in 3 cm wide, 1 cm high aluminium cups. Multiple identical odour cups were used such that an animal never investigated the same cup twice. The odours used included nutmeg, vanillin, coriander, banana, garlic, cinnamon, thyme, almond, onion, curry, ginger, savoury, cumin, dill, jasmine, coffee, oregano, sage, and rosemary. The odours presented and the order of their presentation between animals was pseudorandomized. For habituation, mice were given 15 min of exploration time for each unique context prior to initial exposure. Each exposure was 5 min in length and inter-trial intervals were 15 min. All tests were video-recorded at 60 fps using a NIKON D5200 equipped with a 30 mm lens. An additional overhead video was recorded using a Logitech webcam. All videos were subsequently scored blind to the treatment groups. Exploration was strictly defined as head up sniffing, directed towards and within 1 cm of the odour source. This definition excludes the use of the odour cup for sitting or as support during rearing.

Olfactory spatial memory test: In this paradigm adapted from Eacott and Norman[53], mice were tested for memory of odour location in context. The test chamber was altered to produce two distinct contexts. Zebra-patterned paper was

used to line the walls of context A, whereas context B had transparent walls surrounded by red plastic cups and bedding on the floor. Each animal underwent two encoding phases and one retrieval phase. During the first encoding phase, mice explored context A where two highly distinct odours were placed at opposite ends of the chamber (odour 1 on the left and odour 2 on the right). Next, mice were removed from the chamber and placed in a holding cage. The mice were then returned to the chamber, except it was now configured as context B and contained both odours in opposite positions (odour 2 on the left and odour 1 on the right). The animals were allowed to explore both odours in their new positions before being placed back into the holding cage. For the retrieval phase, the chamber was reconfigured as context A but now two copies of one odour were presented on both sides of the chamber. Time spent investigating the odour cups was measured. The novel configuration consists of the familiar odour in a novel position within the original context. The initial context and left/right position of the odour cups were pseudorandomized.

Olfactory temporal order memory test: This paradigm is based on similar tests used previously to measure memory for the temporal order of objects[54,55]. Mice were tested in a transparent chamber with spatial cues kept constant throughout the session. Each animal underwent three encoding phases and one retrieval phase. In the first exposure, mice were placed in the chamber with two copies of one odour presented on opposite sides of the arena. After exploring both copies, the animal was removed from the chamber and placed in a holding cage. This process was repeated two more times using different odours each time. During the retrieval phase the animal was returned to the chamber, but this time earlier and recently explored odours were presented on opposite sides. In this case, the odour explored earlier is more novel given its TD compared with the recently explored odour. The left–right positions of the first and last odours during the test were pseudorandomized. Time spent investigating both odours was measured.

Novel odour recognition test: This test was given 72 h after examining performance on temporal order memory and followed a similar paradigm. The animals underwent two encoding phases where two copies of a unique odour were presented in each. On the retrieval phase, mice were presented with the initially encountered odour and a previously unexplored odour on opposite sides of the chamber. Time spent investigating each odour was measured.

Olfactory episodic memory test: Animals were first habituated to the apparatus, which consisted of a 50 cm × 50 cm × 20 cm transparent plexiglass open field for a 30 min period. The animals were then exposed to two encoding phases and one retrieval phase each separated by a 1 h delay. In the first encoding phase, animals were given 10 min to explore two different odours located at two adjacent corners of the arena. In the second encoding phase, the animals were given an additional 10 min to explore another set of unique odours presented on the opposite adjacent corners. During the retrieval phase, all four odours were presented with the spatial position of one odour from each set exchanged. This presentation results in each odour possessing a unique spatiotemporal configuration—NL/TD, FL/TD, NL/TR, and FL/TR. Successful memory for an integrated (what, when, and where) memory results in a pattern of exploration such that the odour with the NL/TD configuration is preferentially investigated the most while the FL/TR configuration is investigated the least. Time spent investigating all four odours was measured for 5 min in the retrieval phase. Overhead videos were analysed using the ANY-maze software to produce average heat maps of each treatment group's position within the arena.

Context-driven odour recall test: In this paradigm adapted from Mandairon et al.[10], mice were trained to associate a visually distinct context with an odour and subsequently tested for recollection of the odour when exposed to the context alone. The testing apparatus consists of a 50 cm × 30 cm × 20 cm plexiglas cage with colourful visual patterns pasted on the outside of the walls. A wooden applicator with a cotton swab tip was positioned 3 cm from the floor and 5 cm from one end of the chamber. Before introducing mice into the chamber, 100 µL of a pure odourant was applied to the cotton tip. Each mouse was randomly assigned a monomolecular odourant to be trained with among limonene, isoamyl acetate, nonane, and 1-pentanol. Mice were allowed to explore the context and the odorized cotton swab for 30 min per day for 9 consecutive days. On Day 10, mice were once again placed into the cage, however no odour was added to the cotton swab. Investigation time of the cotton swab was measured on day 9 and 10 for the first 5 min of their exposure to the context. Upon failure to detect the expected odour, mice behaviourally expressed memory by spending a greater amount of time investigating the cotton swab compared to their investigation time when the odour was present on Day 9.

**General histology**. After behavioural testing, mice were transcardially perfused with phosphate-buffered saline (PBS, pH 7.4), followed by 4% paraformaldehyde in phosphate buffer. Brain tissue was extracted and postfixed overnight at 4 °C. The brains were then cryoprotected using a 30% sucrose in PBS solution. Coronal 40 µm-thick sections were collected using a cryostat (Leica, Germany). The sections were slide-mounted, counterstained with 4′,6-diamidino-2-phenylindole 135 for 5 min and subsequently coverslipped with Aquamount (Polysciences, Inc., Warrington, PA). Wide-field fluorescent images were captured using a 4 × objective lens on a fluorescent microscope (Olympus, Japan). Confocal images were captured using a ×20 and ×60 objective through a Quorum spinning disk confocal microscope (Zeiss, Germany). Adobe Photoshop CS6 (Adobe Systems,

Incorporated, San Jose, CA) was used to adjust the brightness and contrast of representative sections.

**Immunohistochemistry**. For c-Fos immunostaining, free-floating tissue sections were obtained and washed with PBS with Triton X-100 (PBS-T) and then blocked with normal donkey serum (5 in 0.1% PBS-T) for 1 h. The sections were subsequently incubated with rabbit polyclonal anti-c-Fos antibody (1:1000 in PBS-T; Santa Cruz Biotechnology, California) for 72 h on a nutating mixer at 4 °C. After incubation with the primary antibody the sections were submerged in Alexa Flour 594-conjugated donkey anti-rabbit secondary antibody (1:500 in PBS-T; Invitrogen) for 90 min at room temperature.

**Validation of chemo- and optogenetic inhibition**. To characterize the AON's response patterns we exposed groups of three animals to one of four conditions: homecage only, homecage with an applied odour (100 µL of limonene on a cotton ball), novel context only, or novel context with an applied odour. With the exception of the homecage control group, all animals were put into each condition for a total duration of 30 min.

To validate our chemogenetic and optogenetic manipulations, we prepared two groups of four animals—one group received unilateral CAV2-Cre infusions into the mAON and Cre-responsive hM4D in the vHPC; another received unilateral ArchT infusions into the vHPC and optic fibre implantations in the mAON. Each group was exposed to a novel odour-context pairing for 10 min (reduced from 30 min to prevent prolonged illumination in the optogenetic experiment). hM4D-expressing mice were injected with CNO 15 min prior to exposure whereas ArchT-expressing mice were illuminated with green light throughout the exposure. In the subsequent tissue analysis, the mean density of c-Fos-labelled neurons were compared between each hemisphere.

To examine context-driven AON activity, two groups of three mice were trained on a context-odour association by placing the animals in a context (50 × 30 × 20 cm plexiglas closed-lid cage with colourful visual patterns pasted on the outer walls) with an applied odour for 1 h over three consecutive days. The mice were killed the following day after exposure to the context-odour pair or the context alone for 30 min. c-Fos density was measured and compared to a third group trained and exposed to a context in the absence of an applied odour.

All animals were transcardially perfused 75 min after initial exposure to their testing condition.

**Cell counting**. Every other section from bregma + 3.20 to + 2.34 was collected and immunostained for c-Fos. Sections were mounted and scanned using a 4 × objective lens on a fluorescent microscope. c-Fos-positive cell counting was performed using the cellSens software (Olympus). The area of each AON section was outlined to form a region of interest (ROI) and the number of mCherry-expressing cells were counted. Calibration parameters were established using randomly chosen prominent neurons and adjusted by raising the threshold of detection to a conservative level, excluding the majority of false positives. Once set, the calibration was kept constant. The density of neurons was calculated by dividing the number of labelled cells in the ROI by the area for each AON section. The mean density was determined for each animal and for all animals in each experimental group.

**Calculations and statistical analysis**. The discrimination ratios were derived from the exploration time of odour-context pairings during the retrieval phase of each test. For all tests based on the spontaneous novelty preference paradigm, the discrimination ratio was calculated as the difference between the times spent exploring the novel and familiar odour-context configurations divided by the total amount of time investigating both odours. Here, a value of zero indicates that the animal investigated both odours to an equal extent. Positive values up to one indicate preference for the novel combination, whereas a negative value indicates greater investigation of the familiar odour-context pair. The discrimination ratios of each group obtained using the behavioural paradigms detailed in Fig. 1 and Fig. 2 were compared to zero (chance performance) using a (two-tailed) one-sample *t*-test.

For the episodic memory test, percent investigation time was calculated by dividing the amount of time spent investigating an individual odour-spatiotemporal configuration by the time investigating all odours and multiplied by 100%:

$$\frac{NL/TD}{(NL/TD + NL/TR) + (FL/TD + FL/TR)} \times 100\%$$

For analysing spatial memory, the difference in the amount of time spent investigating odours with novel and familiar positions was divided by the total investigation time:

$$\frac{(NL/TD + NL/TR) - (FL/TD + FL/TR)}{(NL/TD + NL/TR) + (FL/TD + FL/TR)}$$

For temporal memory, the difference in the time spent investigating odours experienced earlier and later was divided by the total investigation time:

$$\frac{(NL/TD + FL/TD) - (NL/TR + FL/TR)}{(NL/TD + NL/TR) + (FL/TD + FL/TR)}$$

Performance in novelty preference-based paradigms was compared using $t$-test in hM4D experiments and a one-way analysis of variance (ANOVA) with testing group as the factor in optogenetic experiments. Percent investigation time data collected in the episodic memory test was analysed using a two-way ANOVA with testing group and odour-spatiotemporal configuration as factors. A two-way ANOVA was used to analyse data in the contextually cued odour recall test for hM4D and optogenetic experiments, respectively, using experimental day and treatment group as factors. Where appropriate, Tukey's multiple comparisons test was used for post hoc comparisons. Significance was defined as $*P < 0.05$, $**P < 0.005$, $***P < 0.0005$, $****P < 0.0001$.

**Data availability**. All relevant data are available from the corresponding authors upon reasonable request.

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

## Acknowledgements

We thank Drs. Rutsuko Ito, Boyer Winters, Kaori Takehara, Paul Whissell, and John Yeomans for their valuable input regarding the collection and interpretation of data. This research was funded by operating grants to J.C.K. from the Canadian Institutes for Health Research (CIHR) (MOP 496401) and the Natural Sciences and Engineering Council of Canada (NSERC) (MOP 491009).

## Author contributions

A.J.A. and J.C.K. carried out the study conceptualization and experimental design. A.J.A. performed and analysed behavioural experiments. A.J.A. and J.C.K. wrote the manuscript.

## Additional information

**Competing interests:** The authors declare no competing interests.

