## [Peer Review File · Nature Communications]

Reviewers' comments:

Reviewer #1 (Remarks to the Author):

The paper by Aqrabawi and Kim investigates the role of hippocampal projections to the anterior olfactory nucleus in odor-driven behaviors that rely on an apparent episodic memory. They perform loss of function experiments that demonstrate that these spatially organized projections are required for spatial and temporal odor memory. Further, they suggest that this hippocampal projection to the AON is sufficient to trigger the memory of an odor when other contextual cues are present. Overall, the findings are interesting and well done. Indeed, these findings provide novel insight into the elusive role of the AON in olfactory processing. I do have the following concerns.

1) The authors use chemogenetic and optogenetic manipulations, but do not show any evidence that the techniques are working. Although, these are well-established techniques and they observe changes in behavior, they should determine the efficacy of these techniques under their experimental conditions.

2) The experiments in Figure 3 are a bit complicated and difficult to interpret. It is unclear what the predictions are for the manipulations and it would be helpful if the authors could elaborate on the motivation for doing these experiments and state the hypothesized results. Moreover, it is unclear what we learn from this experiment beyond what is learned in the earlier figures.

3) The experiments in Figure 4 address an interesting idea, that context can drive odor memory in the absence of the odor. However, I question whether the behavioral assay can adequately address this. Does the assay really measure context-driven odor memory? In this assay, they present an odor on a swab in a context over nine days and on the tenth day present the mouse with a blank swab in the familiar context. The control animals investigate the blank swab more than the odor swab on the previous day. One possibility is that they investigate the swab because they remember the odor, as the authors claim. Alternatively, they investigate it as a novel object, which has nothing to do with remembrance of the odor. This may be controlled for in Fig 4b where they introduce the blank swab into a novel context. However, it is unclear what mice should do in this context. Perhaps they don't investigate the swab because they are too busy investigating the new context. The interpretation of this experiment is complicated. If the authors showed how the behavior evolves from day 1 it might improve the interpretation of these experiments.

4) Related to Figure 4 the authors imply that the hippocampal to AON projection activates neurons in the AON upon exposure to the context minus the odor. Directly showing this activation would strengthen the impact of the paper and add support to the idea that context can drive odor memory. This could be done with an electrode, or more simply with c-Fos as a proxy for neuronal activity.

Reviewer #2 (Remarks to the Author):

This is an interesting and carefully executed set of studies. They are well conceived and clearly described. The findings are important for people who are interested either the hippocampus or the olfactory system. Although there has been some work done on the organization and connections of the anterior olfactory nucleus (we have known for decades, for example, that it receives a hippocampal input) not much is known about its function (other than its role in odor localization and hints it is involved in odor learning). Perhaps that is because we have lacked the tools to do the kinds of studies presented here until now. I have only minor comments.

1) The "gold standard" definition of pars medialis of the AON was offered by Haberly and Price. The region examined in this work is referred to the medial AON, which actually encompasses a lot of what is usually referred to as pars dorsalis. The authors should devote a sentence or two going over these distinctions to help the readers who might want to follow other work in the AON

2) The lab's present and earlier work indicate that there is a substantial hippocampal input to pars lateralis. The figures are convincing! However, these results also differ from the existing literature... they should point this out 1) to take more credit for it and b) help readers understand that this is a revision in our understanding of the circuitry of the regions. The iHPC injections obviously label the tenia tecta. Is it possible that secondary connections from this region underlie some of the behavioral results?

3) very small changes: a) Please report the sex of the mice used in these studies. b) line 343, "phosphate-buffered saline" need not be capitalized.

Reviewer #3 (Remarks to the Author):

The study examined the role of hippocampal (HPC) projection to the anterior olfactory nucleus (AON) on contextually-cued odour recall, using firstly a spatial odour and a temporal odour task, and subsequently an what-where-when task. The results indicated that inactivation of the HPC-AON pathway, using a chemogenetic approach, during retrieval impaired performance of both the spatial and temporal odour tasks, while leaving performance of a novel object recognition task intact. As there are anatomically distinct HPC-AON pathways from the intermediate HPC to the lateral AON, and ventral HPC to the medial AON, the next experiment using optogenetic inactivation to investigate the roles of these two parallel pathways. Inactivation of the iHPC-IAON impaired spatial odour memory, but not temporal odour memory or novel odour recognition memory while inactivation of the vHPC-mAON pathway impaired temporal but not spatial memory or novel odour recognition. The same pattern of deficits was revealed when the animals were tested in an episodic-like what-where-when task.

These results were taken as evidence of two parallel HPC-AON pathways which separately process spatial and temporal order information. That the results generalise across the tasks (i.e the episodic and separate spatial/temporal tasks) gives weight to the authors conclusions. Of especial note is that the intermediate HPC has a clear role in processing information concerned with the location of odours, while the dorsal HPC has received the most attention in this regard.

Indeed the mis-match time and place cells have been recorded mostly if not exclusively in the dorsal HPC, do the authors believe that this property extend along the longitudinal axis of the HPC. The implications of the functional segregation of within the hippocampus should be discussed in more detail.

There was quite a lot of within group variability in performance, which is not uncommon in spontaneous recognition memory tasks, but it would be useful for the authors to test whether each group of animals could significantly discriminate between the objects in each task by comparing the discrimination ratio of each condition to zero-chance performance. In particular is the lack of effect in the iHPC-IAON inactivation on temporal order memory reliable?

Please find uploaded our revision of the manuscript NCOMMS-18-01210 by Aqrabawi and Kim entitled “*Spatiotemporal information is differentially conveyed by hippocampal projections to the anterior olfactory nucleus during episodic-like odour memory*”. We are most appreciative of the reviewers’ thoughtful and thorough comments, their overall positive consensus view, and the chance to address each comment and thus improve our submission. Specifically, the editor asked us to provide additional data to address the concerns regarding evidence for optogenetic and chemogenetic manipulations, behavioural control to address alternative explanation for the context memory task, as well as evidence supporting activation of AON neurons during the same behavioural task. Below, we respond point-by-point to review #1, #2, and #3.

Review #1

“The paper by Aqrabawi and Kim investigates the role of hippocampal projections to the anterior olfactory nucleus in odor-driven behaviors that rely on an apparent episodic memory. They perform loss of function experiments that demonstrate that these spatially organized projections are required for spatial and temporal odor memory. Further, they suggest that this hippocampal projection to the AON is sufficient to trigger the memory of an odor when other contextual cues are present. **Overall, the findings are interesting and well done. Indeed, these findings provide novel insight into the elusive role of the AON in olfactory processing. I do have the following concerns.**”

Point 1) *The authors use chemogenetic and optogenetic manipulations, but do not show any evidence that the techniques are working. Although, these are well-established techniques and they observe changes in behavior, they should determine the efficacy of these techniques under their experimental conditions.*

We agree with the reviewer that the efficacy of the neuronal manipulations need to be determined, especially in the context relevant to the behavioural paradigms we used. The revised manuscript addresses this concern by including two new results:

First, we examined c-Fos immunoreactivity in the AON after mice were exposed to the following four conditions (Supplementary Figure 1): 1) home cage (a familiar environment), 2) a novel odour stimulus in home cage, 3) a novel context alone without odour stimulus, 4) a novel odour stimulus in a novel context. We found that the number of c-fos positive cells increased dramatically when mice encountered a novel odour stimulus in a novel context. Neither novel odour nor novel context alone was sufficient to increase AON activity above what was observed in the homecage condition.

Second, we examined the efficacy of chemogenetic and optogenetic inhibition. For chemogenetic inhibition, mice received a unilateral infusion with CAV2-Cre and Cre-responsive hM4D into the AON and HPC, respectively. AON activity was then quantified in CNO-treated mice following exposure to a novel odour-context pairing. A significantly lower density of c-Fos positive neurons was observed in the AON ipsilateral to the infusion site in comparison to the contralateral AON (Supplementary Figure 2). For optogenetic inhibition, mice were unilaterally infused with AAV-CaMKIIa-ArchT-eYFP into the vHPC with an optic fiber implanted over the mAON and then exposed to a novel context-odour pairing under green light illumination. ArchT-mediated inhibition of vHPC terminals resulted in a significant

decrease in the density of c-Fos positive neurons in the mAON ipsilateral to the AAV-injection site in comparison to the contralateral mAON (Supplementary Figure 5).

Together, these results demonstrate that our chemogenetic and optogenetic inhibition efficiently suppressed the increase in AON activity induced by a novel odour-context pairing.

Point 2) *The experiments in Figure 3 are a bit complicated and difficult to interpret. It is unclear what the predictions are for the manipulations and it would be helpful if the authors could elaborate on the motivation for doing these experiments and state the hypothesized results. Moreover, it is unclear what we learn from this experiment beyond what is learned in the earlier figures.*

We appreciate the reviewer's thoughtful comment. In previous experiments (Figure 1 and 2), the spatial and temporal aspects of olfactory memory were separately tested in two independent tasks. Under more natural conditions, however, spatial locations and temporal sequences of events are not separated but integrated into a single episodic memory. To this end, we adopted a new behavioural paradigm where the retrieval of spatial and temporal information can be tested simultaneously within a single session.

We updated the manuscript to reflect this reasoning by adding the following texts:

“The behavioural tests employed thus far were designed to separately examine the spatial and temporal elements of episodic odour memory. However, given the unified nature of recollected episodes, it is necessary to probe memory-based behavioural judgements while the animal concurrently weighs both parameters”

We were also motivated to confirm our findings that vHPC-mAON and iHPC-IAON pathways play differentiated roles in olfactory memory recall. However, we feel that this point is better left implicit to improve readability.

Point 3) *The experiments in Figure 4 address an interesting idea, that context can drive odor memory in the absence of the odor. However, I question whether the behavioral assay can adequately address this. Does the assay really measure context-driven odor memory? In this assay, they present an odor on a swab in a context over nine days and on the tenth day present the mouse with a blank swab in the familiar context. The control animals investigate the blank swab more than the odor swab on the previous day. One possibility is that they investigate the swab because they remember the odor, as the authors claim. Alternatively, they investigate it as a novel object, which has nothing to do with remembrance of the odor. This may be controlled for in Fig 4b where they introduce the blank swab into a novel context. However, it is unclear what mice should do in this context. Perhaps they don't investigate the swab because they are too busy investigating the new context. The interpretation of this experiment is complicated. **If the authors showed how the behavior evolves from day 1 it might improve the interpretation of these experiments.***

The reviewer raises an important point here that we have considered carefully. The context-driven odour recall paradigm (Figure 4) was first introduced by Mandairon et al (2014; PubMed ID:24808838). The authors showed that the association of a visual context with an odour allowed the context alone to elicit a behavioral response indicative of odour memory.

Importantly, the behavioural response was accompanied by a specific neural activation pattern in the olfactory bulb that was highly overlapped with that evoked by the associated odour, but not other odours. The authors went on to show that a novel context failed to elicit either behavioral or neural activation. Together, these data lend strong support to the idea that the lack of investigation found in our novel context control is not driven by a distraction, but likely by a missing odor percept.

We fully acknowledge that it is extremely challenging to experimentally disprove that the mice were not distracted by the novel context. Nonetheless, we agree with the reviewer that presenting how the behavior evolves from day 1 will help readers to interpret these results. The revised manuscript now includes a new figure (Supplementary figure 6) showing the progression of odour investigation behaviour from day 1 to day 9.

Point 4) *Related to Figure 4 the authors imply that the hippocampal to AON projection activates neurons in the AON upon exposure to the context minus the odor. Directly showing this activation would strengthen the impact of the paper and add support to the idea that context can drive odor memory. This could be done with an electrode, or more simply with c-Fos as a proxy for neuronal activity.*

We greatly appreciate this suggestion. The revised manuscript now includes a new figure (Supplementary figure 7) to address this concern. In brief, mice were trained on a context-odour association for 3 days and sacrificed the next day following exposure to the context-odour pair or the context alone. Both groups exhibited greater densities of c-Fos positive neurons within the AON compared to mice trained and tested with no applied odour. This context-driven activation is consistent with what was previously reported in other olfactory structures such as piriform cortex and olfactory bulb by Mandairon et al (2014; PubMed ID:24808838) and Gottfried et al (2004; PubMed ID: 15157428).

Review #2

“This is an interesting and carefully executed set of studies. They are well conceived and clearly described. The findings are important for people who are interested either the hippocampus or the olfactory system. Although there has been some work done on the organization and connections of the anterior olfactory nucleus (we have known for decades, for example, that it receives a hippocampal input) not much is known about its function (other than its role in odor localization and hints it is involved in odor learning). Perhaps that is because we have lacked the tools to do the kinds of studies presented here until now. I have only minor comments.”

Point 1) *The “gold standard” definition of pars medialis of the AON was offered by Haberly and Price. The region examined in this work is referred to the medial AON, which actually encompasses a lot of what is usually referred to as pars dorsalis. The authors should devote a sentence or two going over these distinctions to help the readers who might want to follow other work in the AON.*

We thank the reviewer for this insightful suggestion. Revised manuscript now includes the following texts.

“The ring of cells which constitute the AON is typically partitioned into four subregions based on their relative cardinal positions: pars medialis (mAON), pars dorsalis (dAON), pars lateralis (lAON), and pars ventralis (vAON). Although cytoarchitectural, morphological, neurochemical, and connectivity studies each provide an individual basis for defining AON subregional boundaries, the divisions are often arbitrarily drawn (for a review, see Brunjes et al., 2005). ... Thus, for our purposes the mAON and lAON are defined as the regions receiving vHPC and iHPC inputs, respectively”

Point 2) *The lab’s present and earlier work indicate that there is a substantial hippocampal input to pars lateralis. The figures are convincing! However, these results also differ from the existing literature... they should point this out 1) to take more credit for it and b) help readers understand that this is a revision in our understanding of the circuitry of the regions.*

We appreciate the positive comments regarding our present and earlier work. We have updated the manuscript to point out the novelty of our findings with regards to the presence of HPC inputs to the lateral AON as well as their topographic organization.

The iHPC injections obviously label the tenia tecta. Is it possible that secondary connections from this region underlie some of the behavioral results?

We appreciate the reviewer’s careful attention. It remains to be determined whether iHPC-derived signals in the tenia tecta represent synaptic terminals or passing fibers. Furthermore, optic fiber tips implanted over the lateral AON were positioned approximately 1 mm lateral to the tenia tecta. Based on our estimation using the light transmission calculator (<https://web.stanford.edu/group/dlab/cgi-bin/graph/chart.php>), the light intensity at the both dorsal and ventral tenia tecta is approximately 0.4% of original intensity; this is probably an overestimation because light propagates from fiber tip more in the ventral direction than in the lateral direction (see Yizhar et al. 2011 for more detail). While we cannot completely exclude the possibility, we believe that the effect of inhibiting iHPC terminals at the tenia tecta is likely overshadowed by the main effect of inhibiting iHPC-lAON pathway. Surprisingly little is known about the behavioural function of the tenia tecta, and the HPC-tenia tecta pathway is an important area for future research.

Point 3) *very small changes: a) Please report the sex of the mice used in these studies. b) line 343, “phosphate-buffered saline” need not be capitalized.*
The errors have now been corrected.

Reviewer #3

“That the results generalize across the tasks (i.e the episodic and separate spatial /temporal tasks) gives weight to the authors conclusions. Of special note is that the intermediate HPC has a clear role in processing information concerned with the location of odours, while the dorsal HPC has received the most attention in this regard.

Point 1) *Indeed the mis-match time and place cells have been recorded mostly if not exclusively in the dorsal HPC, do the authors believe that this property extend along the longitudinal axis of the HPC. The implications of the functional segregation of within the hippocampus should be discussed in more detail.*

While the HPC is traditionally segregated into dorsal and ventral parts, each specialized in the type of information being processed (dorsal HPC is involved in spatial memory whereas the ventral HPC is more important for emotional memory), new evidence suggest a more complex functional organization for the HPC. This changing perspective is particularly driven by findings such as those described in our manuscript and by others that can not be accommodated from a functional dichotomy viewpoint.

We agree with the reviewer that it is important to further comment on this point in our discussion. Thus we have added the following text:

“While the dorsal hippocampus is commonly thought to dominate spatial processing, particularly when contrasted to the ventral aspect, this dichotomic perspective is being revised in light of growing evidence for a hippocampal model characterized by multiple levels of functional long-axis gradients and discrete domains. This alternative perspective better accommodates recent findings demonstrating iHPC and vHPC involvement in transmitting spatial information and is in accordance with our results”

Point 2) There was quite a lot of within group variability in performance, which is not uncommon in spontaneous recognition memory tasks, but it would be useful for the authors to test whether each group of animals could significantly discriminate between the objects in each task by comparing the discrimination ratio of each condition to zero- chance performance. In particular is the lack of effect in the iHPC-IAON inactivation on temporal order memory reliable?

We thank the reviewer for their helpful comment. We agree that the availability of this data can facilitate interpretation and strengthen the manuscripts conclusions. Indeed, a one-sample t-test comparing the iHPC-IAON group mean discrimination ratio to zero (chance performance) does reveal a significant effect in the temporal order memory paradigm. All groups in each condition detailed in Fig. 1 and Fig. 2 were examined using the same statistical analysis and the results can be found as Supplementary Table 1 and 2.

REVIEWERS' COMMENTS:

Reviewer #1 (Remarks to the Author):

The revised manuscript is improved and the authors have adequately addressed all of my concerns. The manuscript is now appropriate for publication in Nature Communications.

Reviewer #2 (Remarks to the Author):

Reviewer 2

Point 1) Your addition..

"The ring of cells which constitute the AON is typically partitioned into four subregions based on their relative cardinal positions: pars medialis (mAON), pars dorsalis (dAON), pars lateralis (lAON), and pars ventralis (vAON). Although cytoarchitectural, morphological, neurochemical, and connectivity studies each provide an individual basis for defining AON subregional boundaries, the divisions are often arbitrarily drawn (for a review, see Brunjes et al., 2005). ... Thus, for our purposes the mAON and lAON are defined as the regions receiving vHPC and iHPC inputs, respectively"

ignores the fact that Haberly and Price found very different connectivity and cytoarchitecture between pars lateralis, dorsalis, medialis and ventroposterior. Other labs may have divided up the region on the basis of "cardinal points", but they showed serious differences in organization (which is why it was the "gold standard"). I think you need to consider how your maps match up with theirs to make a clear story.

Point 2) The tenia tecta (also known as the "anterior hippocampal rudiment)... is an actual extension of the real hippocampus into the olfactory peduncle. It needs some formal attention as such in the paper. It is there, it lights up at least a bit, it might have some role...

Reviewer #3 (Remarks to the Author):

The authors have addressed the concerns I raised in my original review, and importantly have added additional data which verifies the optogenetic and pharmacogenetic techniques used to manipulate the hippocampal AON pathways.

The authors have also provided additional statistical analyses of the behavioural tasks.

I have no other concerns with the manuscript

Response to reviewer #2

Point 1) *“The ring of cells which constitute the AON is typically partitioned into four subregions based on their relative cardinal positions: pars medialis (mAON), pars dorsalis (dAON), pars lateralis (lAON), and pars ventralis (vAON). Although cytoarchitectural, morphological, neurochemical, and connectivity studies each provide an individual basis for defining AON subregional boundaries, the divisions are often arbitrarily drawn (for a review, see Brunjes et al., 2005). ... Thus, for our purposes the mAON and lAON are defined as the regions receiving vHPC and iHPC inputs, respectively” ignores the fact that Haberly and Price found very different connectivity and cytoarchitecture between pars lateralis, dorsalis, medialis and ventroposterior. Other labs may have divided up the region on the basis of “cardinal points”, but they showed serious differences in organization (which is why it was the “gold standard”). **I think you need to consider how your maps match up with theirs to make a clear story.***

Thank you for the insightful suggestion. We have updated our manuscript by adding the following texts.

Previous studies have found that each subregion displays distinct cytoarchitectural, morphological, neurochemical, and connectivity patterns, thereby providing critical foundations for defining AON subregional boundaries (for a review, see Brunjes et al., 2005). Our recent neuroanatomical study revealed a previously unknown topographic gradient in HPC-AON projections such that the ventral-most part of the HPC innervates most heavily the medial aspect of the AON, and progressively more dorsal parts of the HPC innervate increasingly more lateral positions at the AON. As a result, our definition of the mAON encompasses the pars medialis, yet includes a portion of what is traditionally accepted to be pars dorsalis.

Point 2) *The tenia tecta (also known as the “anterior hippocampal rudiment”)... is an actual extension of the real hippocampus into the olfactory peduncle. **It needs some formal attention as such in the paper.** It is there, it lights up at least a bit, it might have some role...*

Thank you for the insightful suggestion. We have updated our manuscript by adding the following texts.

‘Lastly, in addition to the AON, HPC axon fibers were found, albeit to a much lesser extent, within the tenia tecta, an extension of the HPC into the olfactory peduncle. Little is known about the function of this structure, thus the contribution of HPC inputs to the tenia tecta remains undetermined.’